**DOI: 10.1038/ncomms15412**　　**OPEN**

# Impact of simultaneous exposure to arboviruses on infection and transmission by *Aedes aegypti* mosquitoes

Claudia Rückert[1], James Weger-Lucarelli[1], Selene M. Garcia-Luna[1], Michael C. Young[1], Alex D. Byas[1], Reyes A. Murrieta[1], Joseph R. Fauver[1] & Gregory D. Ebel[1]

The recent emergence of both chikungunya and Zika viruses in the Americas has significantly expanded their distribution and has thus increased the possibility that individuals may become infected by more than one *Aedes aegypti*-borne virus at a time. Recent clinical data support an increase in the frequency of coinfection in human patients, raising the likelihood that mosquitoes could be exposed to multiple arboviruses during one feeding episode. The impact of coinfection on the ability of relevant vector species to transmit any of these viruses (that is, their vector competence) has not been determined. Thus, we here expose *Ae. aegypti* mosquitoes to chikungunya, dengue-2 or Zika viruses, both individually and as double and triple infections. Our results show that these mosquitoes can be infected with and can transmit all combinations of these viruses simultaneously. Importantly, infection, dissemination and transmission rates in mosquitoes are only mildly affected by coinfection.

[1] Department of Microbiology, Immunology, and Pathology, Colorado State University, Fort Collins, Colorado 80523, USA. Correspondence and requests for materials should be addressed to G.D.E. (email: gregory.ebel@colostate.edu).

Chikungunya (CHIKV; genus *Alphavirus*), dengue (DENV; genus *Flavivirus*) and Zika (ZIKV; genus *Flavivirus*) viruses are mainly transmitted to humans by *Aedes aegypti* mosquitoes. As a result, their geographic distributions largely overlap (Fig. 1). As these agents continue to emerge into new regions[1–3], the likelihood of coinfection by multiple *Ae. aegypti*-borne viruses may be increasing. Importantly, the frequency of coinfection in nature and its clinical and epidemiologic implications are poorly understood. The first report of CHIKV and DENV-2 coinfection occurred in 1967 (ref. 4). More recently, coinfections have been reported during various CHIKV/DENV outbreaks, as well as during the recent outbreak of ZIKV in the Americas[5–9]. A relatively high abundance of CHIKV/DENV-coinfected patients (2–26.3%) was found in a number of studies, including outbreaks in Madagascar[10], Gabon[11] and Saint Martin[12]. Furthermore, in a 2016 clinical study in Nicaragua, 27% of all arbovirus-infected patients were viremic for multiple agents, including all possible combinations of CHIKV, DENV and ZIKV[5]. Overall, these studies suggest that coinfections may be fairly common in endemic and epidemic regions.

Coinfected, viremic patients can expose mosquitoes to multiple viruses at the same time. The ability of *Ae. aegypti* mosquitoes to be coinfected and co-transmit arboviruses could have important implications for the epidemiology and evolution of these viruses. However, our current understanding of coinfection and co-transmission by *Aedes* mosquitoes is limited. Notably, during an outbreak of CHIKV and DENV-2 in Gabon in 2010, a CHIKV/DENV-2 coinfected *Aedes albopictus* mosquito was collected around houses of coinfected patients[11]. One laboratory study suggested that *Ae. aegypti* may be refractory to CHIKV/DENV coinfection[13], but two others have reported the ability of *Aedes* mosquitoes to be CHIKV/DENV coinfected and to expectorate both viruses in their saliva[14,15]. Due to a relatively small sample size it was not possible to determine what effect coinfection may have on vector competence. It is not clear whether coinfection with multiple viruses could result in interspecific competition and interference during various stages of infection, or whether infection with one virus could enhance transmission of another. There are several molecular mechanisms supporting the notion that coinfection may influence vector competence. For example, mosquito antiviral responses such as RNA interference (RNAi), among others[16,17], could be activated or suppressed by one of the coinfecting viruses and could thereby indirectly affect replication of another virus. Mechanisms for RNAi suppression in flaviviruses include subgenomic flavivirus RNA[18,19] and an RNAi suppressor function of NS4B (ref. 20). In addition, flavivirus NS1 has recently been shown to be important for initial midgut infection by suppression of immune-related gene expression[21]. Consequently, NS1 may not only enhance flavivirus infection but it may also enhance midgut infection of a heterologous virus such as CHIKV. In contrast, CHIKV replicates and disseminates faster than the two flaviviruses[22–25] and may thus directly outcompete DENV and ZIKV *in vivo*. Similarly, exposing mosquitoes to two closely related flaviviruses, DENV and ZIKV, could directly impact virus infection, dissemination and transmission through superinfection exclusion[26–28] or a similar mechanism. We thus hypothesized that infection of mosquitoes with multiple arboviruses would alter the vector competence for at least one of them.

Accordingly, we sought to determine whether *Ae. aegypti* mosquitoes are capable of transmitting CHIKV, DENV-2 and ZIKV simultaneously as combinations of two or all three viruses. We also sought to quantify how exposure to more than one arbovirus may affect vector competence for the individual viruses. In particular, mosquitoes were exposed to CHIKV, DENV-2 and ZIKV individually, in pairs, and in a triple-infected bloodmeal, and vector competence and virus replication was assessed after 3, 7 and 14 days incubation in mosquitoes. We found that coinfection and co-transmission of all virus pairs by *Ae. aegypti* was possible and occurred frequently. After triple exposure to CHIKV/DENV-2/ZIKV, nearly all mosquitoes became infected by all three viruses, and some of these mosquitoes secreted infectious CHIKV, DENV-2 and ZIKV with their saliva. Collectively, our results indicate that *Ae. aegypti* are capable of co-transmitting all virus pairs as well as the three viruses together, and that coinfection minimally impacts vector competence.

## Results

**Ae. aegypti mosquitoes are susceptible to coinfection.** To maximize infection of mosquitoes and more closely model natural transmission, virus was propagated freshly for each experiment to avoid freeze-thaw and clinically relevant titres were used[29,30]. To estimate virus titres at the time of infection, virus growth curves on Vero cells were established for all three viruses used in this study: CHIKV (strain 99659)[31], DENV-2 (strain Merida)[32] and ZIKV (strain PRVABC59)[33]. Viral genome equivalents (GEs) per ml were determined by quantitative reverse transcriptase PCR (qRT-PCR) (Fig. 2a), and viral plaque forming units (PFU) were determined by plaque assay (Fig. 2b). From these data, the GE/PFU ratio for each time point was determined (Fig. 2c). Peak titres with the lowest GE/PFU ratio were chosen for the collection of fresh virus for experimental infection of mosquitoes (1, 5 and 4 days for CHIKV, DENV-2 and ZIKV, respectively). The workflow for experimental infections/coinfections is shown in Fig. 2d.

We first determined infection rates for mosquitoes exposed to the three viruses individually or in combination. Bloodmeal titres from fresh virus varied between experiments (Table 1) and ranged from $3 \times 10^3$ to $1.4 \times 10^6$ PFU per ml, representative of clinical viremia[5,30]. The overall per cent of mosquitoes that became infected after exposure to single viruses were 87, 81 and 48 for CHIKV, DENV-2 and ZIKV, respectively. At low bloodmeal titres ($\leq 2.2 \times 10^5$ PFU per ml) ZIKV infection rates dropped as low as 5%. We then compared infection rates in singly exposed mosquitoes to those mosquitoes co-exposed to another arbovirus. When mosquitoes were co-exposed to CHIKV/DENV-2, CHIKV and DENV-2 infection rates were not significantly different compared to infection rates after CHIKV or DENV-2 single exposure (Fig. 3a). However, after ZIKV/CHIKV co-exposure, ZIKV infection rates were reduced by 11.3% ($P = 0.0473$, Fisher's exact test) compared to ZIKV single exposure (Fig. 3b), while CHIKV infection rates were unaffected by ZIKV co-exposure. Finally, neither DENV-2 nor ZIKV infection rates were significantly affected by DENV-2/ZIKV co-exposure compared to single exposure (Fig. 3c). High proportions of coinfection with both viruses were observed after co-exposure with all three virus pairings (Fig. 3a–c).

**Dissemination and transmission during coinfection.** Once a mosquito is infected, the virus disseminates through the mosquito body and replicates in the salivary gland before reaching the saliva to be transmitted. To determine whether coinfection affects virus dissemination and transmission, we assessed infection in mosquito legs and salivary secretions collected at all time points among singly infected and coinfected mosquitoes (Fig. 4). Dissemination rates were not significantly affected by coinfection in any of the combinations tested (Fig. 4a,c,e), with the exception of DENV-2 dissemination which was reduced ($P = 0.0149$, Fisher's exact test) at 7 days post infection (d.p.i.) with

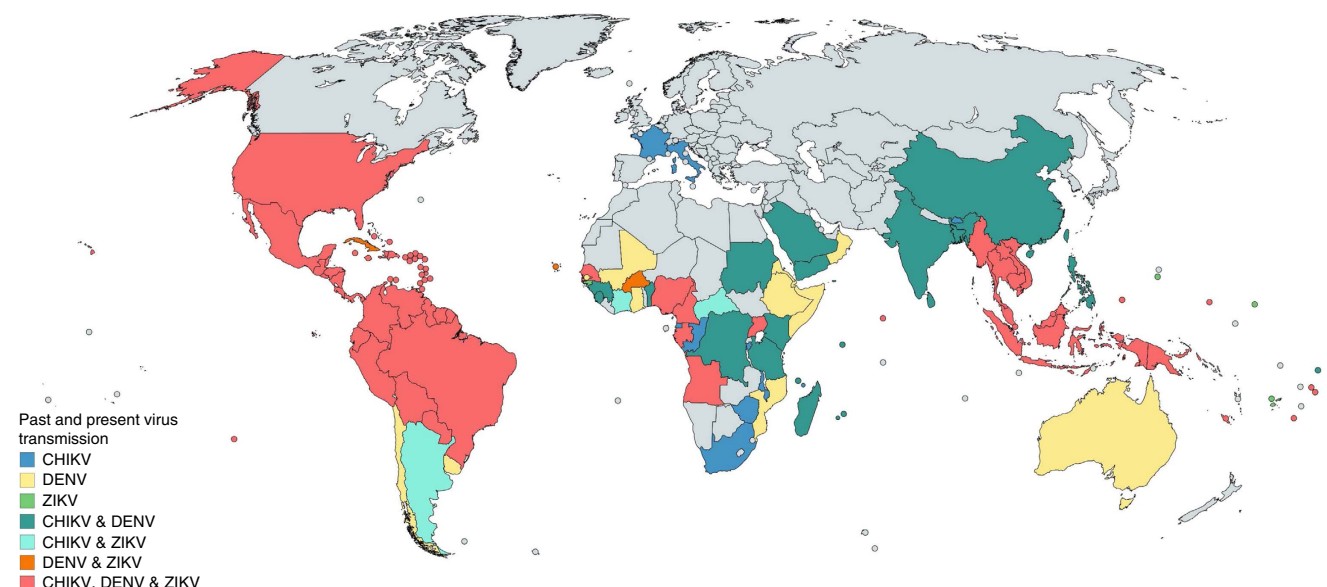

**Figure 1 | Global distribution of three arboviruses.** This map depicts countries with past or current autochthonous transmission of CHIKV, DENV and ZIKV. It was generated using the free online tool https://mapchart.net/detworld.html and is based on data provided by the CDC, PAHO, WHO, the National Institute for Communicable Disease (NICD-NHLS), as well as a review of literature[43-48]. Other countries are also at risk and may have had unreported cases.

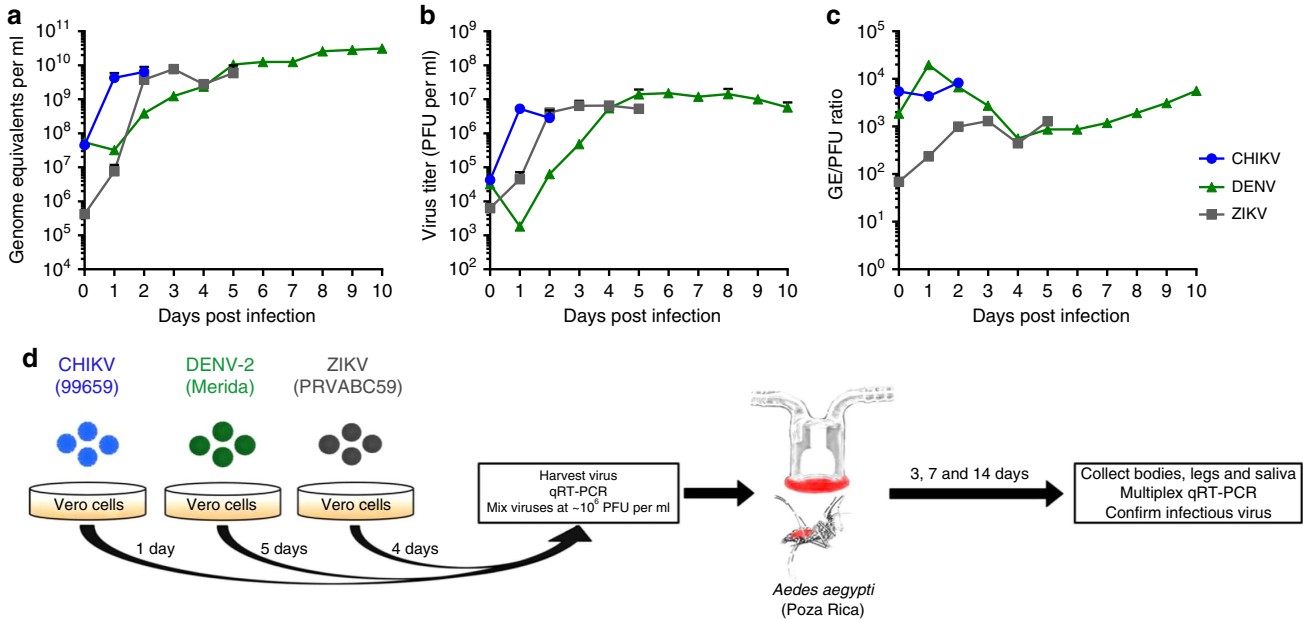

**Figure 2 | Virus replication kinetics and experimental design.** Virus growth curves were used to determine GE per ml (**a**), PFU per ml (**b**) and GE/PFU ratio (**c**) after infection of Vero cells with CHIKV (blue), DENV-2 (black) and ZIKV (green). Arrows indicate the time points used to collect fresh virus stocks for bloodmeals. Data points represent the mean of five biological replicates; error bars represent s.d. The experimental design of mosquito infections/coinfections is shown in **d**.

CHIKV/DENV-2 compared to DENV-2 alone (Fig. 4c). DENV-2 transmission rates remained unaffected by CHIKV or ZIKV coinfection (Fig. 4d), and CHIKV/ZIKV coinfection did not affect transmission rates of either virus (Fig. 4b,f). However, CHIKV transmission rates were significantly reduced ($P = 0.0004$, Fisher's exact test) 14 d.p.i. in DENV-2 coinfected mosquitoes by 27% compared to mosquitoes that were only exposed to CHIKV (Fig. 4b).

Overall, we identified 14, 11 and seven mosquitoes with a co-transmission potential for CHIKV/DENV-2, CHIKV/ZIKV and DENV-2/ZIKV, respectively (Table 2). One saliva sample was positive for CHIKV and ZIKV as early as 3 d.p.i. which corresponded with high virus replication for CHIKV ($1.2 \times 10^8$ RNA copies per sample) and ZIKV ($2.5 \times 10^7$ RNA copies per sample) detected in the legs of the same mosquito compared to other leg samples at 3 d.p.i. (Supplementary Fig. 2a,g). The proportion of co-transmitting mosquitoes was close to, but generally slightly above, expected values given the proportions of mosquitoes that transmit each virus individually (Table 2). Saliva samples positive for two viruses were further screened for infectious virus by inoculation of Vero and/or A549 cells and subsequent measurement of viral GE in the supernatant over time

**Table 1 | Bloodmeal titres and mosquito numbers.**

| Experiment # | Conditions | Virus titre (PFU per ml blood) | n (3 d.p.i.) | n (7 d.p.i.) | n (14 d.p.i.) | n (21 d.p.i.) |
|---|---|---|---|---|---|---|
| 1 | CHIKV | $1.9 \times 10^5$ | 40 | 40 | 40 | NA |
|  | ZIKV | $1.2 \times 10^6$ | 40 | 37 | 24 | NA |
|  | CHIKV/ZIKV | As single* | 40 | 40 | 40 | NA |
| 2 | CHIKV | $1.4 \times 10^6$ | NA | NA | 39 | NA |
|  | ZIKV | $2.2 \times 10^5$ | NA | NA | 40 | NA |
|  | CHIKV/ZIKV | As single* | NA | NA | 40 | NA |
| 3 | CHIKV | $9.7 \times 10^4$ | NA | 40 | 36 | NA |
|  | DENV-2 | $7.4 \times 10^5$ | NA | 40 | 29 | NA |
|  | ZIKV | $1.7 \times 10^4$ | NA | 40 | 40 | NA |
|  | CHIKV/DENV-2 | As single* | NA | 40 | 40 | NA |
|  | DENV-2/ZIKV | As single* | NA | 40 | 40 | NA |
| 4 | CHIKV | $5.6 \times 10^5$ | 40 | 40 | 40 | NA |
|  | DENV-2 | $2.1 \times 10^5$ | 40 | 40 | 40 | NA |
|  | ZIKV | $4.3 \times 10^5$ | 40 | 40 | 40 | NA |
|  | CHIKV/DENV-2 | As single* | 40 | 40 | 40 | NA |
|  | CHIKV/ZIKV | As single* | 40 | 40 | 40 | NA |
|  | DENV-2/ZIKV | As single* | 40 | 40 | 40 | NA |
| 5 | CHIKV | $3.1 \times 10^4$ | 25 | NA | NA | NA |
|  | DENV-2 | $3.0 \times 10^3$ | 32 | NA | NA | NA |
|  | ZIKV | $5.4 \times 10^5$ | 20 | NA | NA | NA |
|  | CHIKV/DENV-2 | As single* | 22 | NA | NA | NA |
|  | DENV-2/ZIKV | As single* | 38 | NA | NA | NA |
| Total | CHIKV | — | 105 | 120 | 155 | NA |
|  | DENV-2 | — | 72 | 80 | 80 | NA |
|  | ZIKV | — | 100 | 117 | 144 | NA |
|  | CHIKV/DENV-2 | — | 62 | 80 | 80 | NA |
|  | CHIKV/ZIKV | — | 80 | 80 | 120 | NA |
|  | DENV-2/ZIKV | — | 78 | 80 | 80 | NA |
| Triple infection† | CHIKV/DENV-2/ZIKV | CHIKV: $3.3 \times 10^6$ DENV-2: $3.6 \times 10^5$ ZIKV: $3.6 \times 10^6$ | NA | NA | 48 | 48 |

NA, not applicable.
*Dual infections contained the same titres for each virus as single infections.
†The triple infection was performed as a separate experiment and was analysed separately.

(Supplementary Table 1). Selected single positive samples and negative saliva samples were included as controls. We identified seven, six and five saliva samples containing infectious CHIKV/DENV-2, CHIKV/ZIKV and DENV-2/ZIKV, respectively, including the CHIKV/ZIKV positive saliva sample from 3 d.p.i. In total, 28 out of 37 selected samples that were RT-PCR positive for one virus tested positive for infectious virus, and all RT-PCR-negative samples were negative for infectious virus.

**Viral load in saliva of single and dual infected mosquitoes.** To assess the impact of coinfection on virus replication dynamics within mosquitoes, we compared viral GEs in the different tissues of single infected and coinfected mosquitoes. We found a number of small but statistically significant changes in GE in legs and bodies during coinfection/co-exposure compared to single exposure (Supplementary Figs 2 and 3). However, none of these effects were greater than threefold and are not reflected in changes in dissemination or transmission. Importantly, the viral load in saliva was comparable in singly infected, co-exposed and co-transmitting mosquitoes for all viruses both at 7 d.p.i. (Fig. 5a,c,e) and 14 d.p.i. (Fig. 5b,d,f). The geometric mean RNA load in single infections at 7 d.p.i. was $5.0 \times 10^3$ GE per saliva sample for CHIKV (Fig. 5a), $3.0 \times 10^3$ GE per DENV-2 (Fig. 5c) and $1.3 \times 10^3$ GE per saliva sample for ZIKV (Fig. 5e). At 14 d.p.i. the geometric means were $5.3 \times 10^3$ GE per saliva sample for CHIKV (Fig. 5b), $8.2 \times 10^3$ GE per saliva sample for DENV-2 (Fig. 5d) and $3.5 \times 10^3$ GE per saliva sample for ZIKV (Fig. 5f). No significant differences were observed between GE in saliva samples from single exposed, dual exposed or co-transmitting mosquitoes.

**Ae. aegypti can transmit all three viruses simultaneously.** Finally, we exposed 48 mosquitoes to all three viruses and collected legs, saliva and bodies at 14 and 21 days post exposure (d.p.e.) and screened them for viral RNA. In total, 92% of mosquitoes were infected with all three viruses at both time points combined (Table 3). One mosquito was uninfected and seven mosquitoes were positive only for CHIKV and ZIKV. All but one triple-infected mosquito had established a disseminated infection at 14 d.p.e.; all had a disseminated infection at 21 d.p.e. Six saliva samples were PCR positive for all three viruses at 14 d.p.e. and two at 21 d.p.e. In addition, seven saliva samples were positive for two of the three viruses. Saliva samples which tested PCR positive for more than one virus were screened for infectious virus by inoculation of Vero cells and/or A549 cells (Supplementary Table 1). All three viruses were infectious to Vero and/or A549 cells in four out of the six triple positive saliva samples collected at 14 d.p.e. The other two samples were positive for two viruses (one CHIKV + ZIKV and one DENV-2 + ZIKV). Out of the two 21 d.p.e. triple positive samples, one was positive for ZIKV and the other for DENV-2 and ZIKV.

The triple infection was not performed in parallel to the dual/single infections and may thus not allow for an in-depth comparison (due to variability in bloodmeal titres). It is still noteworthy that the overall CHIKV transmission rate at 14 d.p.i. was 26% lower ($P = 0.0016$, Fisher's exact test) than in singly infected mosquitoes, which is comparable to the reduction in CHIKV transmission after CHIKV/DENV-2 coinfection. However, the geometric mean of CHIKV RNA ($2.2 \times 10^4$ GE per saliva) in these samples was fourfold higher compared to CHIKV singly infected mosquitoes ($5.3 \times 10^3$ GE per saliva). Viral loads of DENV-2

$(5.8 \times 10^3$ GE per saliva) were comparable to singly infected mosquitoes $(8.2 \times 10^3$ GE per saliva), but ZIKV RNA loads $(8.1 \times 10^4$ GE per saliva) were 23-fold increased compared to singly infected mosquitoes $(3.5 \times 10^3$ GE per saliva). Interestingly, none of the saliva samples were positive for infectious CHIKV at 21 d.p.i. (Table 3, Supplementary Table 1).

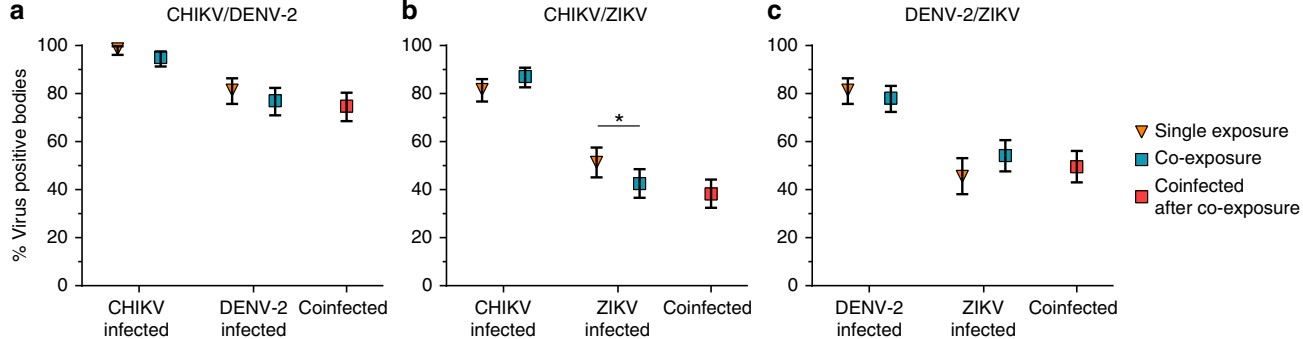

**Figure 3 | Susceptibility of *Aedes aegypti* to arbovirus infection after single and dual exposure.** *Ae. aegypti* mosquitoes were exposed to CHIKV, DENV-2 or ZIKV either individually or in combination by infectious bloodmeal. Infection rates for individual viruses after single exposure (yellow triangle) with CHIKV (**a,b**), DENV-2 (**a,c**) or ZIKV (**b,c**), and parallel co-exposures (blue square) with CHIKV/DENV-2 (**a**), CHIKV/ZIKV (**b**) and DENV-2/ZIKV (**c**) are presented. These values represent infection rates for each virus independent of coinfection status. Coinfection rates (that is, mosquitoes positive for both viruses) after co-exposure are presented separately (red square). Since there was no significant difference in the infection rate between time points (Supplementary Fig. 1), the combined proportion (%) from all replicates and time points are presented with 95% confidence interval (error bars). Total mosquito numbers were as follows: (**a**) CHIKV ($n = 221$), DENV-2 ($n = 221$), CHIKV/DENV-2 ($n = 222$); (**b**) CHIKV ($n = 279$), ZIKV ($n = 261$), CHIKV/ZIKV ($n = 280$); and (**c**) DENV-2 ($n = 221$), ZIKV ($n = 180$), DENV-2/ZIKV ($n = 238$). Statistical significance was determined by Fisher's exact test (*$P < 0.05$).

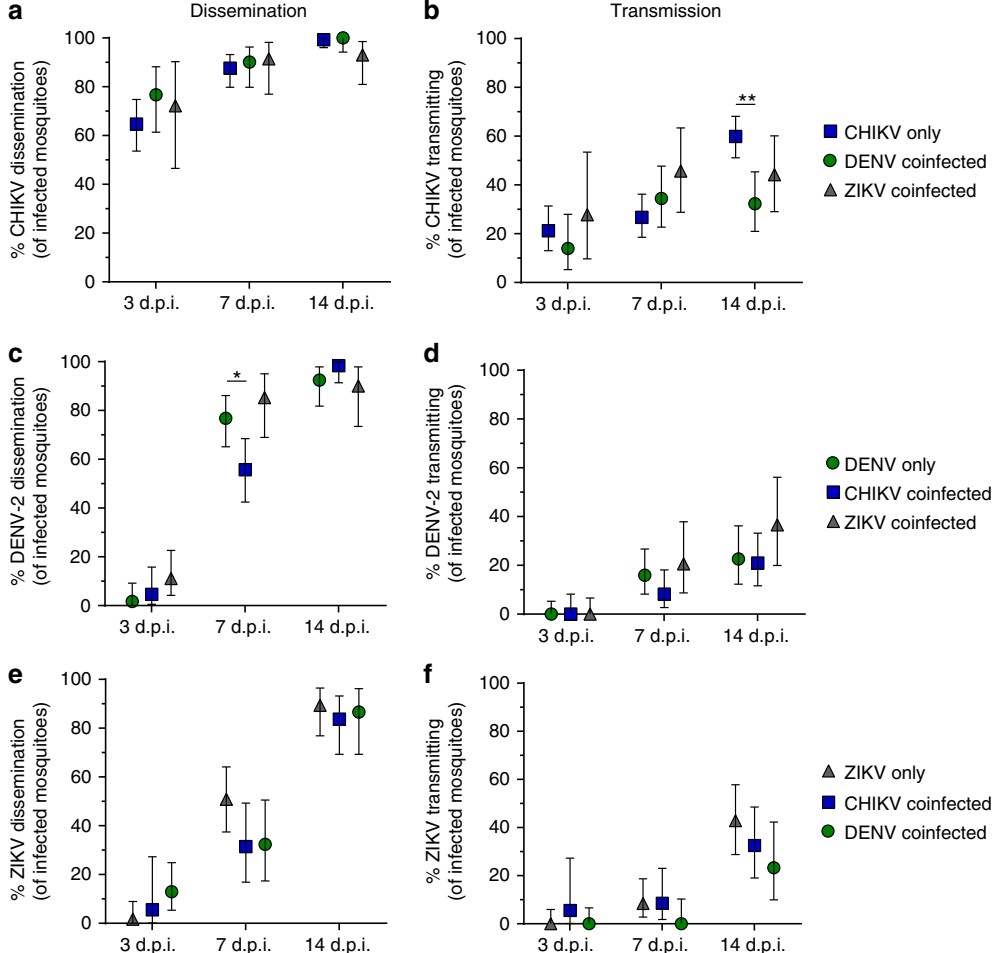

**Figure 4 | Dissemination and transmission dynamics in single and double infections.** Dissemination rates (**a,c,e**) and transmission rates (**b,d,f**) for CHIKV (**a,b**), DENV-2 (**c,d**) and ZIKV (**e,f**) in single and double infected mosquitoes. The overall percentage and 95% confidence interval from two independent experiments is shown. Statistical significance as indicated (*$P < 0.05$; **$P < 0.01$) was calculated using Fisher's exact test.

**Table 2 | Virus transmission rates in single and dual infections.**

| Exposure | 3 d.p.i. observed | 3 d.p.i. predicted* | 7 d.p.i. observed | 7 d.p.i. predicted* | 14 d.p.i. observed | 14 d.p.i. predicted* | Total |
|---|---|---|---|---|---|---|---|
| CHIKV only† | 18/105 (17.1%) | — | 28/120 (23.3%) | — | 82/160 (51.3%) | — | 128/385 (33.2%) |
| DENV-2 only† | 0/72 (0%) | — | 11/80 (13.8%) | — | 12/69 (17.4%) | — | 23/221 (10.4%) |
| ZIKV only† | 0/100 (0%) | — | 5/117 (4.3%) | — | 21/144 (14.6%) | — | 26/361 (7.2%) |
| CHIKV + DENV-2‡ | 0/62 (0%) | 0% | 5/80 (6.3%) | 3.2% | 9/80 (11.3%) | 8.9% | 14/222 (6.3%) |
| CHIKV + ZIKV‡ | 1/80 (1.3%) | 0% | 3/80 (3.8%) | 1.0% | 7/120 (5.8%) | 7.5% | 11/380 (2.9%) |
| DENV-2 + ZIKV‡ | 0/78 (0%) | 0% | 0/80 (0%) | 0.6% | 7/80 (8.8%) | 2.5% | 7/232 (3%) |

*Predicted probability of co-transmission was calculated by multiplication of single transmission proportions.
†Ae. aegypti exposed to viruses individually; numbers indicate transmission potential for one virus.
‡Ae. aegypti exposed to two viruses; numbers indicate transmission potential for both viruses simultaneously.

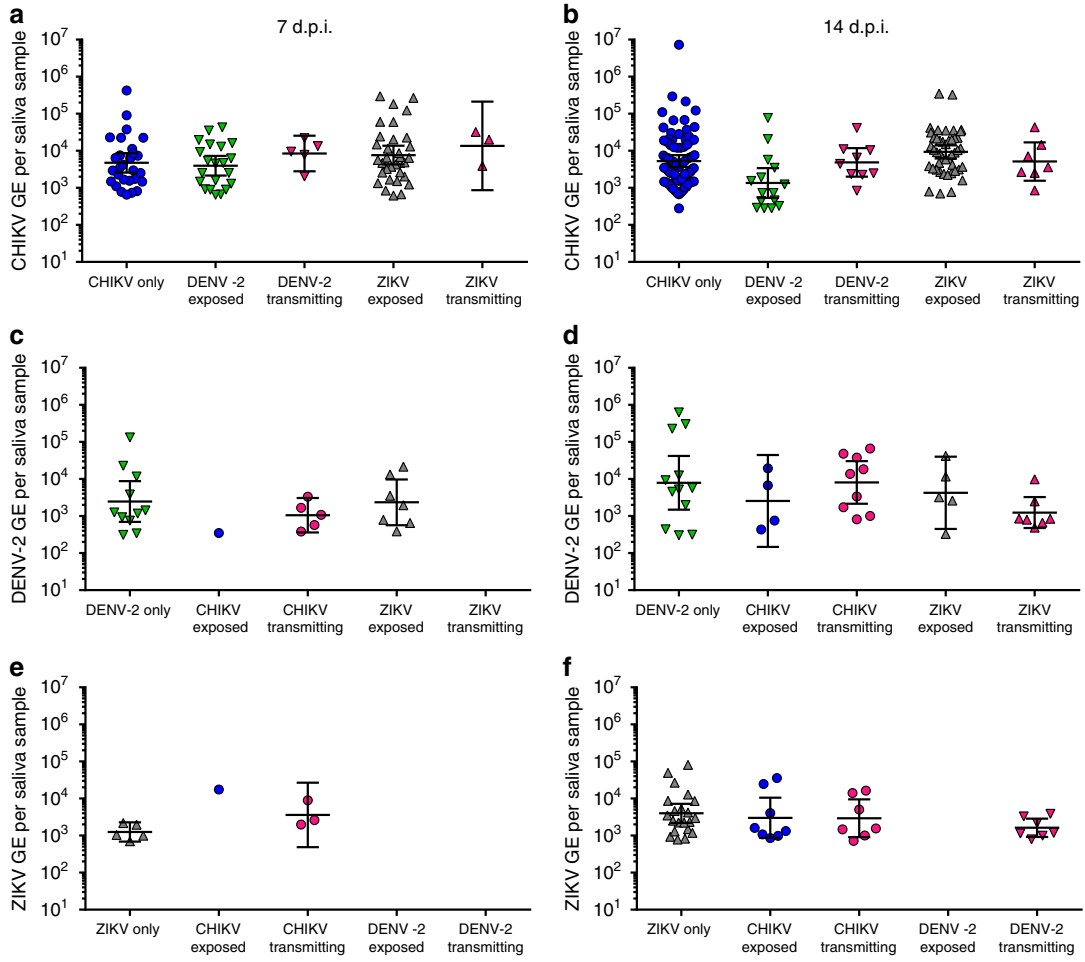

**Figure 5 | RNA levels in saliva of single and coinfected Aedes aegypti.** GE/saliva are shown for samples positive for CHIKV (**a,b**), DENV-2 (**c,d**) and ZIKV (**e,f**) at both 7 d.p.i. (**a,c,e**) and 14 d.p.i. (**b,d,f**). The geometric mean is indicated and error bars represent the 95% confidence interval. Statistical significance was determined using one-way ANOVA with multiple comparisons.

## Discussion

Recent field and clinical data have indicated that in nature, human beings and mosquitoes may be infected by multiple arboviruses more frequently than has been previously appreciated[6–9,29,34]. Unfortunately, the extent to which coinfecting viruses interfere or synergize with one another *in vivo* is poorly understood. Therefore, we sought to assess coinfection of *Ae. aegypti* mosquitoes with more than one arbovirus. We used an *Ae. aegypti* colony of Mexican origin and virus isolates from the Americas to mimic the current outbreak in the Americas; however, vector competence for different arboviruses in various

*Ae. aegypti* populations varies, for example by geographic region, and our study may not be representative of some other mosquito/virus combinations. Our results indicate that *Ae. aegypti* (Poza Rica) mosquitoes exposed to more than one arbovirus become coinfected frequently, even at relatively low titres. In addition, exposure to multiple viruses tended not to affect mosquito susceptibility: mosquito midgut infection rates for each virus were similar whether the virus was delivered alone or in combination with another arbovirus. The reason for this is likely multifactorial. First, it may be that CHIKV, DENV-2 and ZIKV target different cells (or subcellular compartments) within the mosquito midgut.

| Time | Sample | All negative | CHIKV only | DENV only | ZIKV only | CHIKV + DENV positive | CHIKV + ZIKV positive | DENV + ZIKV positive | Triple positive |
|---|---|---|---|---|---|---|---|---|---|
| 14 d.p.i. | Infection* | 1/48 | 0/48 | 0/48 | 0/48 | 0/48 | 2/48 | 0/48 | 45/48 |
| | Dissemination* | 1/48 | 0/48 | 0/48 | 0/48 | 0/48 | 3/48 | 0/48 | 44/48 |
| | Transmission* | 35/48 | 4/48 | 0/48 | 0/48 | 2/48 | 0/48 | 1/48 | 6/48 |
| 21 d.p.i. | Infection* | 0/48 | 0/48 | 0/48 | 0/48 | 0/48 | 5/48 | 0/48 | 43/48 |
| | Dissemination* | 0/48 | 0/48 | 0/48 | 0/48 | 0/48 | 5/48 | 0/48 | 43/48 |
| | Transmission* | 33/48 | 0/48 | 2/48 | 7/48 | 0/48 | 1/48 | 3/48 | 2/48 |

**Table 3 | Vector competence of mosquitoes simultaneously exposed to all three viruses.**

*Infection, dissemination and transmission potential was determined by viral RNA in bodies, legs and saliva, respectively.

Alternatively (or possibly in addition), it may be that only a small number of cells are initially infected in the midgut[35], rendering enhancement and/or superinfection exclusion during coinfection largely irrelevant. Furthermore, it remains unclear whether superinfection exclusion even occurs between these viruses since the literature is somewhat contradictory[26–28]. In general, our results allow us to conclude that in most cases arbovirus infection of mosquito midguts is not altered by the presence of coinfecting arboviruses. However, we observed one case where co-exposure appeared to minimally impact infection of mosquito midguts. CHIKV/ZIKV co-exposure resulted in decreased ZIKV infection in mosquitoes. While the reduction was statistically significant, it was not dramatic (11.3%). The mechanisms that produced this small effect are unclear at present, but may be related to direct competition. Overall, these data suggest that there is likely little interaction between multiple arboviruses during the establishment of infection in Ae. aegypti.

Following infection of mosquito midguts, arboviruses must disseminate from the midgut and ultimately infect salivary glands and be released into mosquito salivary secretions. Modulated immune responses or direct virus competition for resources may suppress or enhance virus replication and dissemination during coinfection. However, we found that dissemination and transmission rates were similar in single and coinfected mosquitoes. Only CHIKV/DENV-2 coinfection resulted in any changes in dissemination and transmission dynamics, and these changes were small. While CHIKV co-exposure significantly reduced DENV-2 dissemination at 7 d.p.i., transmission rates were not significantly reduced and no difference was observed at 14 d.p.i. Moreover, DENV-2 coinfection significantly reduced CHIKV transmission at 14 d.p.i. by 27%. Notably, this was the biggest effect that coinfection had on vector competence for any of the viruses. Yet taken altogether, these findings indicate how little coinfection affects vector competence and that, for the most part, there appear to be sufficient resources in the mosquito for multiple viruses to replicate systemically. In fact, when we used single transmission proportions to predict how likely co-transmission is, we found that the proportion of mosquitoes that were co-transmitting was either close to the predicted proportion or in some cases higher. This increase was probably related to an overlap in high/low vector competence in individual mosquitoes for these viruses; for example, a mosquito more competent for DENV-2 may also be more competent for ZIKV resulting in co-transmission proportions higher than random probability.

In addition to the co-transmission numbers, viral GE expectorated with saliva were not significantly different in co-exposed or co-transmitting mosquitoes compared to single infections, suggesting that the transmission potential of single versus dual transmitting mosquitoes is comparable. Selected saliva samples were also infectious for more than one virus. While we were not able to detect infectious virus in all PCR-positive saliva samples, this was not unexpected due to different sensitivities of various assays and differences in specimen content and handling. Moreover, our results clearly establish that Ae. aegypti mosquitoes have the capability to transmit more than one virus during a single feeding episode.

Finally, triple infection with and transmission of CHIKV, DENV-2 and ZIKV was confirmed. Triple infection rates were high and co-transmission rates were comparable to single transmission rates of DENV-2 and ZIKV, but lower for CHIKV. The impact of our observed increase in CHIKV and ZIKV GE in triple positive saliva on transmission to naïve hosts is currently unclear. While exposure to the three viruses simultaneously is likely to be an extremely rare occurrence, these results support the overall conclusion from much of the work reported here, that multiple arboviruses can be transmitted by a single mosquito during a single feeding episode.

The implications of co-exposure and co-transmission on the epidemiology, pathogenesis and evolution of these agents remains unclear. Due to limited clinical information and, most likely, underdiagnosis of coinfections, the extent of coinfections in human beings and mosquitoes under natural conditions is poorly defined. Thus, future work will be aimed at understanding the effect of coinfection on pathogenesis in mammals. However, coinfection may also play an important role in virus evolution, both in the mammalian host and mosquito vector. Beyond the possibility that increased or generally altered immune pressures may drive virus evolution during coinfection, it is unclear whether different viruses may replicate within the same cells allowing for recombination events. We thus need to improve our understanding of where in the mosquito these viruses replicate during coinfection compared to single infection and how coinfection affects virus evolution in the mosquito and the mammalian host. Another important question is how coinfection and co-transmission rates may differ when mosquitoes are exposed to multiple viruses at different stages of their adult life—in the present study we expose mosquitoes to virus with their first bloodmeal, but we do not know how mosquito age or previous bloodmeals may affect vector competence during coinfection. Furthermore, sequential exposure to multiple viruses will be another aim of future studies and may have a more pronounced effect on vector competence. However, if coinfection and co-transmission after sequential exposure is also common, it could further increase the number of co-transmitting mosquitoes and the risk of coinfection. In conclusion, this study provides clear evidence that CHIKV, DENV-2 and ZIKV may be co-transmitted by mosquitoes following simultaneous exposure. The impact of coinfection on the biology of the agents and human health are high priorities for future work.

## Methods

**Cells.** Vero cells (ATCC CCL-81) were maintained in Dulbecco's modified Eagle's medium (DMEM) containing 5% fetal bovine serum (FBS) and 50 μg ml$^{-1}$

gentamycin at 37 °C with 5% $CO_2$. A549 cells (ATCC CCL-185; kindly provided by Dr Nisha Duggal, CDC, Fort Collins) were maintained in DMEM 10% FBS and 50 µg ml$^{-1}$ gentamycin at 37 °C with 5% $CO_2$. Cells tested negative for mycoplasma/bacterial contamination by DAPI stain and fluorescence microscopy analysis.

**Mosquitoes.** *Ae. aegypti* (*Stegomyia*) colonies (F12–14) established from wild populations in Poza Rica, Mexico[36], were used for vector competence studies. For colony maintenance, mosquitoes were fed citrated sheep blood and given sugar and water *ad libitum*. Larvae were reared and adults maintained under controlled conditions of temperature (28 °C), humidity (70% RH) and light (12:12L:D diurnal cycle).

**Viruses.** American isolates of CHIKV (R99659; British Virgin Islands; GenBank #KJ451624), DENV-2 (BC-17; Merida, Mexico, GenBank #AY449677) and ZIKV (PRVABC59; Puerto Rico; GenBank #KU501215) were obtained from the Center for Disease Control and Prevention branch in Fort Collins, CO (CHIKV and ZIKV) and Dr William Black IV (DENV-2). All three viruses were propagated in Vero cells by infection at multiplicity of infection (MOI) 0.01 to generate virus stocks which were stored in aliquots at −80 °C. Virus titres (PFU per ml) were quantified using Vero cells and a standard plaque assay protocol.

**Multiplex qRT-PCR.** To screen for the three viruses in one sample, a multiplex qRT-PCR assay was established. Previously published primers and probes were used for CHIK (modified from Grubaugh et al.[37]) and ZIKV (Lanciotti et al.[38]). A new set of primers and probe were designed for DENV-2. All primers are listed in Supplementary Table 2. T7 RNA standards were generated for each PCR using primers as listed in Supplementary Table 2 for CHIKV and DENV-2. For ZIKV standards, RNA was produced from a full-length infectious clone as described previously[39]. Sensitivity and efficiency were highly comparable between single and multiplex assays using the selected primer sets, and no cross-reactivity was observed (Supplementary Fig. 4). When the assay was used to determine positivity of mosquito samples, a conservative Ct value cutoff of 36.5 was used, roughly corresponding to ten RNA copies. RNA from unexposed mosquitoes never amplified above that threshold for any of the three viruses.

**In vitro experiments.** Growth curves were established to determine peak of virus production. For this, five replicate wells of a six-well plate were infected at MOI 0.1 with CHIKV, DENV or ZIKV. At each time point, 200 µl supernatant was taken, centrifuged at 3,000g for 5 min and supernatant was transferred to a new tube. An aliquot of 50 µl was taken and treated with RNaseA (final 200 µg ml$^{-1}$) for 30 min at 37 °C to digest free RNA in the cell supernatant. The samples were immediately lysed in TNA lysis buffer (Omega) and frozen at −20 °C. RNA was extracted from the RNaseA-treated supernatant using the Mag-Bind Viral DNA/RNA 96 kit (Omega Bio-Tek) on the KingFisher Flex Magnetic Particle Processor (Thermo Fisher Scientific). RNA was eluted in 50 µl nuclease-free water. Genome equivalents per ml were determined by qRT-PCR using primers and probes also used for multiplex qRT-PCR as described above (Supplementary Table 2). Virus titres (PFU per ml) were determined by plaque assay using the remaining supernatant from each sample. Growth curves were stopped when >50% of the cells in a culture were dead.

**Coinfection of Aedes aegypti.** Four- to seven-day-old *Ae. aegypti* were orally exposed to blood containing the three viruses individually or in combination using a water-jacketed glass feeder and hog-gut as a membrane. The infectious bloodmeal contained 50% defibrinated calf blood, 10% FBS and 40% DMEM containing freshly propagated virus stocks to result in an estimated titre of 10$^6$ PFU per ml of blood (of each virus in dual or triple infections). CHIKV, DENV-2 and ZIKV were grown on Vero cells for 1, 5 and 4 days, respectively. The mean GE/PFU ratio determined at the selected time points (CHIKV = 4,308; DENV2 = 867; ZIKV = 445) was used to estimate titres of fresh virus stocks prior to mosquito bloodmeal. Stock titres were subsequently confirmed by plaque assay (Table 1) and were in the majority of infections lower than $1 \times 10^6$ PFU per ml. Groups of 40 mosquitoes were fed for each time point and condition wherever possible. Due to variable availability of female mosquitoes taking an infectious bloodmeal, five experiments were performed to result in at least two replicate experiments per condition and time point (Table 1). All mosquito infections were performed under BSL3 conditions. Mosquitoes were dissected at 3, 7 and 14 days to collect bodies, legs and saliva to determine infection, dissemination and transmission rates, respectively. Samples were screened by qRT-PCR for viral RNA. Selected saliva samples were also used to inoculate Vero and/or A549 cells to determine the presence of infectious virus in saliva samples.

**Mosquito dissections and sample processing.** Dissections and sample processing were performed as described previously[23]. Briefly, at the selected time points post-infectious bloodmeal, mosquitoes were cold-anaesthetized and legs and wings were dissected into a 2 ml tube containing 250 µl mosquito diluent ($1 \times$ PBS

supplemented with 20% FBS, 50 µg ml$^{-1}$ Penicillin/Streptomycin, 50 µg ml$^{-1}$ Gentamycin and 2.5 µg ml$^{-1}$ Fungizone) and a stainless steel bead. The mosquitoes' proboscis was then placed into a capillary tube filled with immersion oil for 20 min to salivate. Bodies were then also placed into a 2 ml tube containing 200 µl diluent and a stainless steel bead. The end of the capillary tubes containing saliva were broken off into 100 µl diluent and centrifuged at 15,000g for 5 min. Mosquito tissues (legs/wings and bodies) were homogenized using a Retsch Mixer Mill MM400 (Germany) at 25 cycles per second for 1 min and centrifuged at 15,000g for 5 min at 4 °C. After homogenization and centrifugation, 50 µl of each sample was transferred into TNA lysis buffer (Omega) and the remaining sample volume was frozen at −80 °C. RNA was extracted from the lysed sample using the Mag-Bind Viral DNA/RNA 96 kit (Omega Bio-Tek) on the KingFisher Flex Magnetic Particle Processor (Thermo Fisher Scientific). Samples positive for viral RNA were identified by multiplex qRT-PCR.

**Confirmation of infectious virus in saliva.** For selected samples including those which were positive for more than one virus, 10 µl of saliva was used to infect Vero cells in a 24-well format. Supernatant was taken at 0 h as well as 48 h, 72 h and/or 120 h post infection. An increase in viral RNA over time was used to identify samples containing infectious virus. In some samples CHIKV was highly abundant and resulted in >50% cytopathic effect at 48 hours post infection (h.p.i.), making it difficult to determine the presence of infectious DENV-2 or ZIKV which replicate slower. These samples were also used to infect A549 cells which are resistant to CHIKV infection[40], but susceptible to DENV-2 (ref. 41) and ZIKV[42].

**Statistical analysis.** A two-tailed Fisher's exact test was used to compare rates of infection, dissemination and transmission in vector competence studies. One-way ANOVA with multiple comparisons was used to compare GEs in saliva samples. GraphPad Prism 6.0 (La Jolla, CA) was used for statistical tests and significance was defined as $P < 0.05$.

**Global distribution map generation.** Figure 1 was generated using the free online tool https://mapchart.net/detworld.html and is based predominantly on data provided by the CDC, WHO, PAHO, the National Institute for Communicable Disease (NICD-NHLS), as well as a review of literature[43–48]. Only countries with past and/or current autochthonous transmission of CHIKV, DENV and ZIKV are shown.

**Data availability.** The authors declare that all relevant data are available within the article file and its Supplementary Information or from the corresponding author upon request.

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

## Acknowledgements

The authors would like to acknowledge William Black IV for providing the DENV-2 strain and Brandy Russel (CDC, Fort Collins) for providing the CHIKV and ZIKV strains used in this study. Furthermore, we would like to acknowledge Nisha Duggal (CDC, Fort Collins) for providing the A549 cells, Daniel Hartman and Nicolas Bergren for assistance with mosquito dissections, as well as Muhammad Mahdi Karim for providing an *Aedes aegypti* image that we modified for use in Fig. 2d. This work was supported by NIH grant AI067380. S.M.G.-L. was supported by the NIH Fogarty grant 2D43TW001130-0681 and R.A.M. was supported by the Colorado State University GAUSSI fellowship 2016-2017 (NSF grant DGE-1450032).

## Author contributions

C.R., J.W.-L. and G.D.E. designed the study. C.R., J.W.-L., S.M.G.-L., M.C.Y., A.D.B., R.A.M. and J.R.F. performed experiments. C.R. analysed the collected data. C.R. and G.D.E. drafted the manuscript. All authors critically revised the manuscript and approved the final version.

## Additional information

**Competing interests:** The authors declare no competing financial interests.

