## [Peer Review File · Nature Communications]

REVIEWERS' COMMENTS:

Reviewer #1 (Remarks to the Author):

This is for the most part a well written paper that addresses an unexplored aspect of arbovirus infection of mosquitoes and transmission, including Zika virus. The experiments are robust and well explained, resulting in high quality data providing novel insights on arbovirus transmission by Aedes. The manuscript is thereby suitable for publication.

The authors should elaborate on possible differences in this phenomenon (the fact that coinfection does not affect infection rates and transmission by individual viruses) that could relate to different Aedes aegypti strains. It seems like the Poza Rica strain is quite susceptible to all three viruses. Several strains have however been shown to differ in their susceptibility to infection with a specific virus, and in this case the results of similar experiments could have been different, and not necessarily predictable; i.e. that the virus which is less effective at infecting in a single infection assay would also show lower titers in a coinfection scenario. The same argument could be made for different virus isolates/serotypes.

The other aspect that needs to be further discussed relates to the timing of virus acquisition, which in nature could vary greatly. For example, what would happen if a mosquito acquires the different viruses at different times during their adult life, and not only those tested in the present study. The sentence on lines 104 - 109, is difficult to understand. The figure 3 legend can also be improved in this respect.

Reviewer #2 (Remarks to the Author):

This paper is well written and should be of interest to a small group in the community. This type of work has been completed recently, so the novelty is not high and results are not unexpected. Although these viruses are circulating in the same areas by the same mosquitoes, the number of triple infections may not be too significant. Overall, this work adds to the field and should be of interest to other researchers in the field.

The tables and figures were fairly easy to interpret except FIG 3: the explanation of the co-exposure vs. coinfection after co-exposure should be expanded.

Minor editorial comments:

line 51 - remove other

The Results section seem to have a lot of methods in it, I am not sure how much of these items could be moved to the methods section.

check for consistency in the use of blood meal vs. bloodmeal - example line 94

line 226 ...between these viruses SINCE the literature....

References should be standardized

Overall a nice paper - my primary concern is the appropriateness of this work for this journal.

Response to the reviewers

Reviewer #1 (Remarks to the Author):

This is for the most part a well written paper that addresses an unexplored aspect of arbovirus infection of mosquitoes and transmission, including Zika virus. The experiments are robust and well explained, resulting in high quality data providing novel insights on arbovirus transmission by Aedes. The manuscript is thereby suitable for publication.

The authors should elaborate on possible differences in this phenomenon (the fact that coinfection does not affect infection rates and transmission by individual viruses) that could relate to different Aedes aegypti strains. It seems like the Poza Rica strain is quite susceptible to all three viruses. Several strains have however been shown to differ in their susceptibility to infection with a specific virus, and in this case the results of similar experiments could have been different, and not necessarily predictable; i.e. that the virus which is less effective at infecting in a single infection assay would also show lower titers in a coinfection scenario. The same argument could be made for different virus isolates/serotypes.

We thank the reviewer for their kind words about our manuscript and taking the time to evaluate it. We agree with the point that different mosquito colonies or virus strains could have a different phenotype – we added a section to the discussion (lines 170-173) to highlight that we are aware of this constraint of the study.

The other aspect that needs to be further discussed relates to the timing of virus acquisition, which in nature could vary greatly. For example, what would happen if a mosquito acquires the different viruses at different times during their adult life, and not only those tested in the present study.

We agree that this is an important point and addressed this aspect in lines (lines 230-234) of our discussion.

The sentence on lines 104 - 109, is difficult to understand. The figure 3 legend can also be improved in this respect.

We made an attempt to improve this section, clarifying what we were trying to say (lines 90-98). We also added further explanation to the Figure 3 legend, which will hopefully make this section overall more understandable.

Reviewer #2 (Remarks to the Author):

This paper is well written and should be of interest to a small group in the community. This type of work has been completed recently, so the novelty is not high and results are not unexpected. Although these viruses are circulating in the same areas by the same mosquitoes, the number of triple infections may not be too significant. Overall, this work adds to the field and should be of interest to other researchers in the field.

We thank the reviewer for their comments and taking the time to review our manuscript. We agree that the observed co-transmission itself may not be particularly unexpected to scientists in the field, since it has been shown that *Aedes albopictus* can co-transmit CHIKV&DENV (Vazeille et al. 2010; Nuckols et al. 2015) and Nuckols et al. also reported one co-transmitting *Aedes aegypti* mosquito.

However, we believe that the scale of our study is the first to allow for a strong comparison of vector competence of *Aedes aegypti* during single and dual infection, which was not possible in previous studies (smaller scale, lower dissemination rates and very little transmission in Nuckols et al.). In the end, we were still surprised that the impact of coinfection on vector competence was not more pronounced, and when discussing our research with other scientists we have had great interest in our data. Furthermore, we were not sure what to expect from coinfection with two related viruses such as ZIKV and DENV and were at least somewhat surprised to see that this coinfection combination had little effect on vector competence.

We agree that triple infection is most likely a particularly rare occurrence and we have added a statement regarding this to the discussion (lines 217-218).

The tables and figures were fairly easy to interpret except FIG 3: the explanation of the co-exposure vs. coinfecting after co-exposure should be expanded.

Thank you, we have expanded our figure legend to clarify and make interpretation of the data easier.

Minor editorial comments:

line 51 - remove other

Thank you for drawing our attention to this – the unnecessary word ‘other’ was removed (line 52).

The Results section seems to have a lot of methods in it, I am not sure how much of these items could be moved to the methods section.

We reduced method descriptions in the results section where possible (line 85; following paragraph was removed) and moved the necessary portions to the method section (lines 297-301).

check for consistency in the use of blood meal vs. bloodmeal - example line 94

We have removed the space in instances where we had used ‘blood meal’ to make the use of bloodmeal consistent.

line 226 ...between these viruses SINCE the literature....

We agree that this is the better wording and have changed ‘and’ to ‘since’ (line 182).

References should be standardized

The reference list has been updated from the ‘Nature’ style to the ‘Nature communications’ style using EndNote, and as far as we can tell is now in line with previous ‘Nature communication’ publication.